# Opioids depress breathing through two small brainstem sites

**Iris Bachmutsky[1,2], Xin Paul Wei[1,3], Eszter Kish[1,2], Kevin Yackle[1]\***

[1]Department of Physiology, University of California-San Francisco, San Francisco, United States; [2]Neuroscience Graduate Program, University of California-San Francisco, San Francisco, United States; [3]Biomedical Sciences Graduate Program, University of California-San Francisco, San Francisco, United States

**Abstract** The rates of opioid overdose in the United States quadrupled between 1999 and 2017, reaching a staggering 130 deaths per day. This health epidemic demands innovative solutions that require uncovering the key brain areas and cell types mediating the cause of overdose— opioid-induced respiratory depression. Here, we identify two primary changes to murine breathing after administering opioids. These changes implicate the brainstem's breathing circuitry which we confirm by locally eliminating the μ-Opioid receptor. We find the critical brain site is the preBötzinger Complex, where the breathing rhythm originates, and use genetic tools to reveal that just 70–140 neurons in this region are responsible for its sensitivity to opioids. Future characterization of these neurons may lead to novel therapies that prevent respiratory depression while sparing analgesia.

## Introduction

Nearly 400,000 people in the United States died from a drug overdose involving a prescription or illicit opioid between 1999 and 2017 (*Scholl et al., 2018*). This epidemic is not unique to the United States and with the increasing distribution of highly potent synthetic opioids like fentanyl, it has become a global public health emergency (*Rudd et al., 2016*). Death from opioid overdose results from slow and shallow breathing, also known as opioid induced respiratory depression (OIRD, *Pattinson, 2008*). Like humans, breathing in mice is severely depressed by opioids and this response is eliminated when the μ-Opioid receptor (*Oprm1*) is globally deleted (*Dahan et al., 2001*). *Oprm1* is broadly expressed, in both the central and peripheral nervous systems, including sites that could modulate breathing such as: the cerebral cortex, brainstem respiratory control centers, primary motor neurons, solitary nucleus, and oxygen sensing afferents (*Mansour et al., 1994*; *Kirby and McQueen, 1986*). Therefore, either one or multiple sites could be mediating the depressive effects of opioids on breathing.

Indeed, multiple brain regions have been shown to independently slow breathing after local injection of opioid agonists (*Kirby and McQueen, 1986*; *Montandon et al., 2011*; *Mustapic et al., 2010*; *Prkic et al., 2012*). Although informative, doubts remain for which of these sites are necessary and sufficient to induce OIRD from systemic opioids for three reasons. First, injection of opioid agonists or antagonists into candidate areas modulates μ-opioid receptors on the cell body (post-synaptic) as well as receptors on incoming terminals (pre-synaptic). Second, these studies necessitate anesthetized and reduced animal preparations, which alter brain activity in many of the candidate *Oprm1* expressing sites. And third, there is not a standard and quantitative definition for how breathing changes in OIRD, and this makes comparing studies that use different breathing metrics measured in different experimental paradigms challenging.

To address these limitations, we conducted a detailed quantitative analysis of OIRD in awake animals and identify two key changes to the breath that drive the depressive effects of opioids. These

**\*For correspondence:**
Kevin.Yackle@ucsf.edu

**Competing interests:** The authors declare that no competing interests exist.

**eLife digest** Opioids such as morphine or fentanyl are powerful substances used to relieve pain in medical settings. However, taken in too high a dose they can depress breathing – in other words, they can lead to slow, shallow breaths that cannot sustain life. In the United States, where the misuse of these drugs has been soaring in the past decades, about 130 people die each day from opioid overdose. Pinpointing the exact brain areas and neurons that opioids act on to depress breathing could help to create safer painkillers that do not have this deadly effect. While previous studies have proposed several brain regions that could be involved, they have not been able to confirm these results, or determine which area plays the biggest role.

Opioids influence the brain of animals (including humans) by attaching to proteins known as opioid receptors that are present at the surface of neurons. Here, Bachmutsky et al. genetically engineered mice that lack these receptors in specific brain regions that control breathing. The animals were then exposed to opioids, and their breathing was closely monitored.

The experiments showed that two small brain areas were responsible for breathing becoming depressed under the influence of opioids. The region with the most critical impact also happens to be where the breathing rhythms originate. There, a small group of 50 to 140 neurons were used by opioids to depress breathing. Crucially, these cells were not necessary for the drugs' ability to relieve pain.

Overall, the work by Bachmutsky et al. highlights a group of neurons whose role in creating breathing rhythms deserves further attention. It also opens the possibility that targeting these neurons would help to create safer painkillers.

two metrics thereafter define OIRD in our study and can serve as a rubric for others. We then locally eliminate the µ-opioid receptor in awake mice, disambiguating pre and post-synaptic effects, and use these metrics to define two key brain sites that mediate OIRD. Recently, a similar approach demonstrated some role for these sites in OIRD (*Varga et al., 2020*). Among these two sites in our study, we find that one is dominant and driven by just 140 critical neurons in vitro and, importantly, these neurons are not required for opioid-induced analgesia, suggesting a neutral target for developing safer opioids or rescue strategies for opioid overdose.

## Results

Up to now, OIRD has generally been described as a slowing and shallowing of breath (*Pattinson, 2008*). We therefore felt it was important to more precisely, quantitatively describe the changes in breathing in hopes of elucidating potential mechanisms of respiratory depression. We began by asking whether specific parameters of the breath are affected by opioids. We monitored breathing in awake, behaving mice by whole body plethysmography after intraperitoneal injection (IP) of saline for control and then 20 mg/kg morphine at least 24 hr later (*Figure 1A*). Compared to saline, breathing after morphine administration (in normoxia) became much slower and inspiratory airflow decreased, each by 60% (*Figure 1B,C*). This culminated in ~50% decrease in overall minute ventilation (MV = approximated tidal volume x respiratory rate, *Figure 1C*), demonstrating that 20 mg/kg morphine is, indeed, a suitable dose to model OIRD.

Breath morphology in normoxia after IP saline versus morphine cannot be directly compared since activity of the mouse is different (exploring vs. sedated, *Supplementary file 1*), which significantly influences the types of breaths taken. This prevented a precise characterization of breath parameters that dictate OIRD. To overcome this, we measured breathing in hypercapnic air (21% $O_2$, 5% $CO_2$) which normalizes behavior and thus breathing (*Figure 1D*, *Supplementary file 1*). As in normoxia, morphine depressed respiratory rate (by 50%, *Figure 1E,F*), peak inspiratory airflow (by 60%, *Figure 1E,F*), and minute ventilation (by 60%, *Figure 1F*). Hypercapnic breaths after saline exhibited two phases, inspiration and expiration, each lasting about 50 msec. (*Figure 1G,H*). After morphine, only the inspiratory phase (measured as inspiratory time, Ti) became substantially longer (*Figure 1G,H*). Additionally, hypercapnic breaths showed a new, third phase after the initial expiration (measured as expiratory time, Te, *Figure 1G,H*) that was characterized by prolonged little to no airflow (<0.5 mL/sec.) preceding hypercapnia induced active expiration (*Pisanski and Pagliardini,*

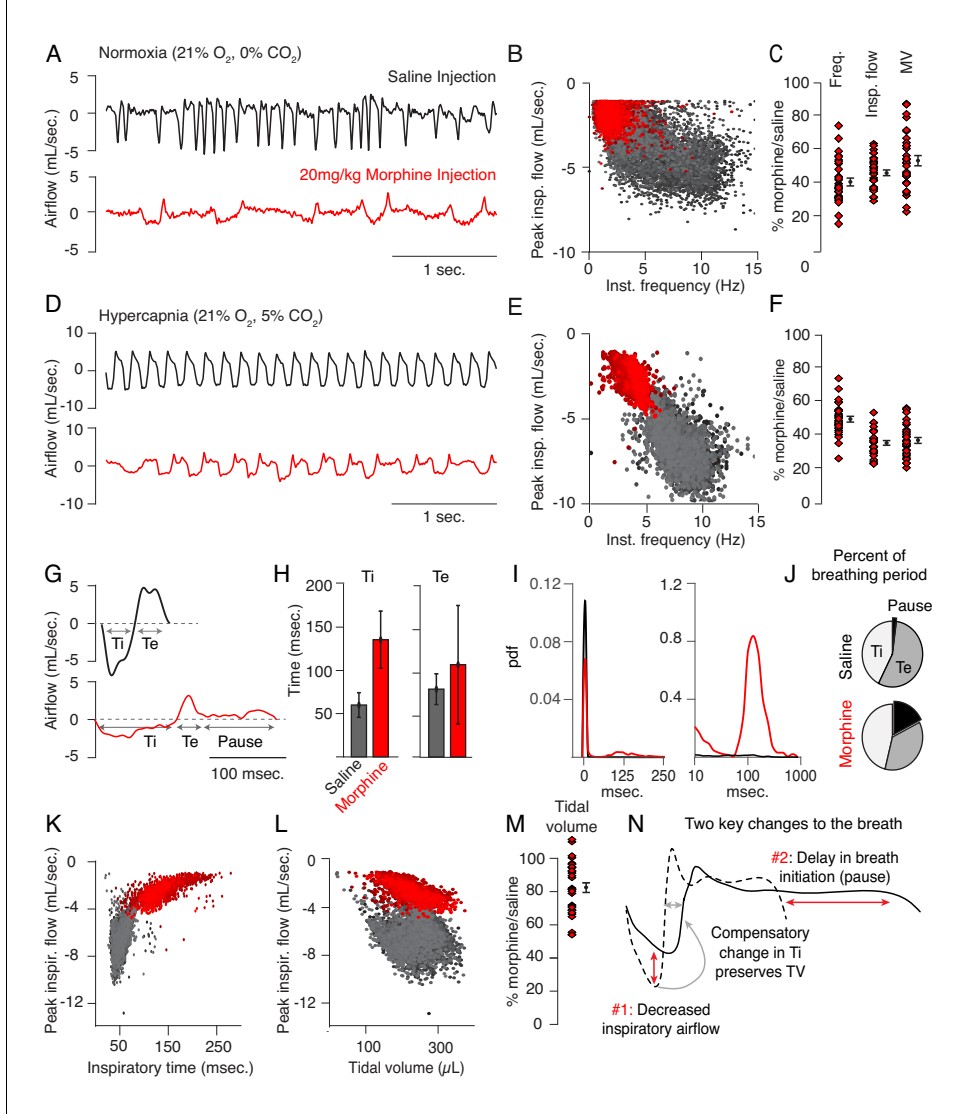

**Figure 1.** Changes to breathing during opioid-induced respiratory depression. (**A**) Representative examples of the breathing airflow (mL/sec.) in normoxia (21% $O_2$) measured by whole body plethysmography. 15 min before recordings, animals are intraperitoneally (IP) injected with either saline (black) or 20 mg/kg morphine (red). Morphine recordings are captured 1–7 days after saline. (**B**) Scatterplot of instantaneous breathing rate (Hz) versus airflow (mL/sec.) for each breath (dot) taken during the 40 min recordings. Morphine causes breathing to become slow and less forceful. (**C**) Ratio of average breathing parameters after IP injection of morphine-to-saline. Respiratory rate, peak inspiratory airflow, and minute ventilation (MV = approximated tidal volume*rate) for n = 29 animals in normoxia. Red diamond, single animal average. Black diamond, average of all animals. Error bar, standard error of mean (SEM). (**D–E**) Representative example of breathing airflow and instantaneous scatter plot (rate vs. airflow) from a 10 min whole body plethysmography recording of breathing in hypercapnia (21% $O_2$, 5% $CO_2$) to minimize changes in breathing due to differences in behavior after morphine injection. (**F**) Ratio of respiratory rate (p-value=$1\times10^{-19}$, Cohen's d = 5.96), peak inspiratory airflow (p-value=$1\times10^{-22}$, Cohen's d = 6.18), and minute ventilation (p-value=$1\times10^{-20}$, Cohen's d = 5.31) after IP injection of morphine-to-saline for n = 29 animals in hypercapnia. (**G**) Representative single breath airflow trace for breaths in hypercapnia after saline (black) or morphine (red) IP injection. Hypercapnic saline breaths can be divided into two phases whose durations (msec.) can be measured: inspiration (Ti) and expiration (Te). Hypercapnic morphine breaths have a third phase after expiration where airflow is nearly 0 mL/sec., which we call a pause. (**H**) Bar graph of the average length ± standard deviation of Ti and Te for a single representative animal. (**I**) Probability density function plot of the pause length in breathing during hypercapnia after saline (black) or morphine (red) IP injection on a numerical (left) and logarithmic scale (right). Note, morphine selectively increases Ti and pause length. (**J**) Percent of the

*Figure 1 continued on next page*

*Figure 1 continued*

average breath period spent in inspiration, expiration, or pause for hypercapnic breaths after saline or morphine injection. (K) Scatterplot of inspiratory time (msec.) vs. peak inspiratory airflow (mL/sec.) for 10 min of hypercapnic breaths after saline (black) or morphine (red) IP injection. As inspiratory time increases, peak inspiratory flow decreases. (L) Scatterplot of tidal volume (μL.) vs. peak inspiratory airflow (mL/sec.) for 10 min of hypercapnic breaths after saline (black) or morphine (red) IP injection. Even though peak inspiratory airflow decreases after morphine, tidal volume is preserved due to prolonged Ti. (M) Ratio of tidal volume after IP injection of morphine-to-saline for n = 29 animals in hypercapnia (p-value=$3\times10^{-6}$, Cohen's d = 1.13). (N) Schematic of the two key morphine induced changes to the breath: decreased inspiratory airflow and pause. Decreased inspiratory airflow prolongs Ti since negative feedback from the lung reflecting breath volume is slower. We interpret the pause as a delay in initiation of the subsequent inspiration.

The online version of this article includes the following source data and figure supplement(s) for figure 1:

**Source data 1.** Average frequency, peak flow, minute ventilation, tidal volume and pause duration for 31 animals after intraperitoneal saline or morphine in normoxia and hypercapnia.
**Figure supplement 1.** Examples of morphine breaths in hypercapnia binned by pause length.
**Figure supplement 2.** Scatter plot of expiratory duration with and without the pause to identify an effective airflow threshold.

---

*2019*). We define this new phase as a pause (low airflow + active expiration, *Figure 1G*, *Figure 1—figure supplement 1*). Such pauses lasted up to several hundred milliseconds (*Figure 1I*), accounting for about one-third of the average breath length (*Figure 1J*). Thus, the 50% decrease in respiratory rate after morphine administration is primarily due to prolonging of Ti and pause phases, and the increased prevalence of time spent in pause significantly contributes to the decrease in minute ventilation.

Typically the length of inspiratory time is determined by a stretch-evoked feedback signal from the lung which terminates inspiration (*West, 2005*). This reflex is represented by the correlation observed between Ti and peak inspiratory airflow (*Figure 1K*). Breaths in morphine still maintain this correlation despite having a longer Ti and decreased inspiratory airflow (*Figure 1K*). As a result, morphine breaths have a similar approximated tidal volume (TV) compared to saline control (*Figure 1L,M*). In other words, as opioids decrease inspiratory airflow, Ti displays a compensatory increase to preserve TV (*Figures 1N* and *Hill et al., 2018*). In summary, opioids cause only two primary changes to the breath, namely, 1) decreased inspiratory airflow and 2) addition of a pause phase that delays initiation of subsequent breaths (*Figure 1N*). These two parameters can both be controlled by the breathing central pattern generator, the preBötzinger Complex (preBötC), in the brainstem and suggest that this may be a key locus affected during OIRD (*Smith et al., 1991*; *Feldman et al., 2013*; *Cui et al., 2016*).

Indeed, the preBötC has been proposed to play a key role in OIRD since localized injection of opioids results in respiratory depression and localized naloxone reverses decreased breathing after administration of systemic opioids (*Montandon et al., 2011*; *Montandon and Horner, 2014*). However, such experiments fail to distinguish between the action of opioids on presynaptic terminals (*Mudge et al., 1979*) of distant neurons projecting into the preBötC versus direct action on preBötC neurons themselves (*Figure 2A*; *Montandon et al., 2011*; *Dobbins and Feldman, 1994*; *Gray et al., 1999*) To overcome this, we genetically eliminated the μ-Opioid receptor (*Oprm1*) from preBötC cells exclusively, sparing projecting inputs, by stereotaxic injection of adeno-associated virus constitutively expressing Cre (AAV-Cre-GFP) into the preBötC of *Oprm1* flox/flox (*Oprm1$^{f/f}$*) adult mice (*Figure 2B*). Injection site specificity was confirmed by the restricted expression of Cre-GFP (*Figure 2—figure supplement 1*), and subsequent *Oprm1* deletion, was inferred. To establish a baseline, we first measured breathing after administration of saline and morphine in normoxia and hypercapnia in intact animals, as described above. At least one month after bilateral injection of virus into the preBötC, we then re-analyzed breathing (*Figure 2C*). With this protocol, each animal's unique breathing and OIRD response serves as its own internal control, which is necessary due to the variability in OIRD severity between mice (*Figure 1F*). Deletion of *Oprm1* in the preBötC did not affect breathing observed after saline injection (*Figure 2D,E*), suggesting that in this context, opioids do not exert an endogenous effect. In contrast, breathing was markedly less depressed by morphine administration (*Figure 2D,E*) compared to the intact control state: breaths

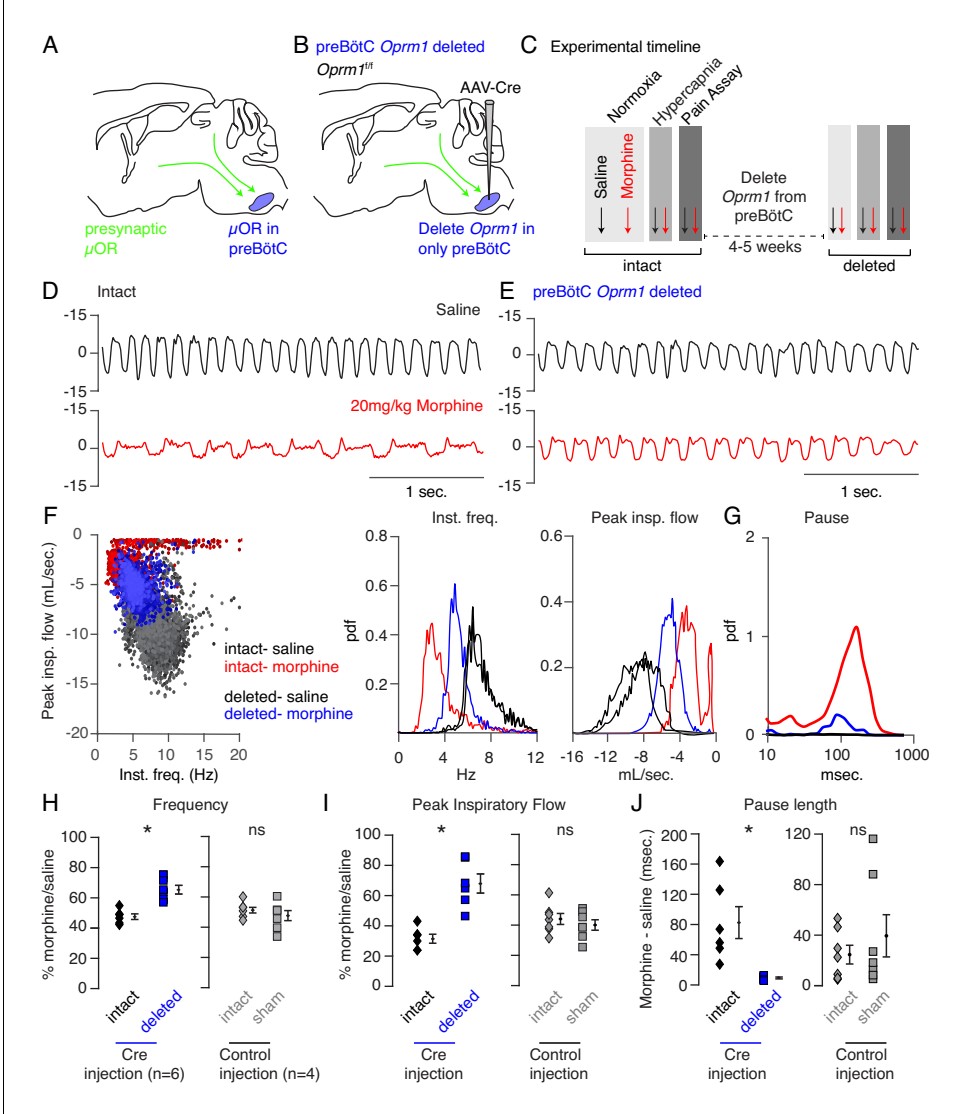

**Figure 2.** Necessity of preBötC ventrolateral brainstem in opioid-induced respiratory depression. (**A**) Schematic of a sagittal section through the adult mouse brain. The μ-Opioid receptor (*Oprm1*) is expressed by a subset of preBötC neurons and is also expressed on presynaptic terminals of some neurons projecting to the preBötC. This confounds the effects observed after localized preBötC injection of opioid or naloxone to investigate its role in OIRD. (**B**) To overcome this, we eliminate *Oprm1* only from the preBötC, and not the presynaptic inputs, to define the role of preBötC neurons in OIRD. (**C**) Experimental time-course. Breathing is measured 15 min after IP injection of saline or 20 mg/kg morphine in both normoxia and hypercapnia. Then *Oprm1*$^{f/f}$ animals are injected with a constitutive-Cre AAV into the preBötC bilaterally. After several weeks breathing is assayed again as above. In this way, each animal's breathing before viral injection can serve as its own control breathing. Response to pain is also measured with a tail-flick assay before and after viral injection to ensure that analgesic response is unaffected. (**D**–**E**) Representative examples of the breathing airflow (mL/sec.) in hypercapnia after saline (black) or morphine (red) before (**D**) and after (**E**) Cre-virus injection. (**F**) Scatter plot of instantaneous respiratory frequency vs. airflow (ml/sec), as well as probability density function plots of both parameters for a representative animal during hypercapnia after saline (black) or morphine (red, blue) IP injection, before (red) and after (blue) Cre-injection. (**G**) Probability density function plot of pause length (msec.) for a representative animal during hypercapnia after IP morphine before (red) and after (blue) Cre-injection, log scale. Prevalence of long duration pauses is greatly reduced. (**H**–**J**) Ratio of average breathing parameters after IP injection of morphine-to-saline. Respiratory rate (**H**) p-value=0.003, Cohen's d = 3.0), peak inspiratory airflow (**I**) p-value=0.0005, Cohen's d = 3.01), and pause length (**J**) p-value=0.04, Cohen's d = 2.0) for 6 *Oprm1*$^{f/f}$ animals with Cre-virus injected into the preBötC or 7 control animals with reporter-virus injected into the preBötC. All sham p-values were not significant (>0.09) and Cohen's D < 0.6. For each experiment 'intact' values are before viral injection, with 'deleted' and 'sham' values

*Figure 2 continued on next page*

*Figure 2 continued*

representing post viral injection conditions in experimental and control animals, respectively. Diamond and square, single animal average. Mixed repeated two-way ANOVA comparing Cre vs. Sham injected was statistically significant for respiratory rate ($F_{(1,11)}=5.5$, p-value=0.04) and pause ($F_{(1,11)}=6.2$, p-value=0.03), but not for peak inspiratory airflow ($F_{(1,11)}=2.1$, p-value=0.17) Black diamond, average of all animals. Error bar, standard error of mean (SEM). * indicates p-value<0.05. ns indicates p-value>0.05.

The online version of this article includes the following source data and figure supplement(s) for figure 2:

**Source data 1.** Average frequency, peak flow and pause duration after intraperitoneal saline or morphine in hypercapnia at baseline or after *Oprm1* deletion from the preBötC.

**Source data 2.** Average frequency, peak flow and pause duration after intraperitoneal saline or morphine in hypercapnia at baseline or after sham viral injection into the preBötC.

**Figure supplement 1.** Expression of GFP protein from AAV-Cre-GFP after bilateral preBötC injection.

**Figure supplement 2.** Necessity of the preBötC for complete opioid induced respiratory depression in normoxia.

**Figure supplement 3.** Tail flick response before and after bilateral preBötC or PBN/KF injection.

---

were twice as fast (3 to 6 Hz, *Figure 2F,H*), the peak inspiratory flow was larger (*Figure 2F,I*), and pauses were nearly eliminated (*Figure 2G,J*). Notably, histological analysis confirmed that AAV-Cre-GFP expression was localized to the preBötC (*Figure 2—figure supplement 1*), and AAV-GFP or tdTomato injected control mice without removal of *Oprm1* showed no change in OIRD compared to the pre-injected control state (*Figure 2H–J*), demonstrating that animals do not develop tolerance to opioids within our experimental timeline. Importantly, rescue of OIRD similarly occurred in normoxia (*Figure 2—figure supplement 2*) and was also specific to breathing since opioids induced analgesia in tail-flick assay after deletion of *Oprm1* in the preBötC (*Figure 2—figure supplement 3*).

Although key features of OIRD (inspiratory airflow and pause) were attenuated by preBötC AAV-Cre injection, rescue was incomplete. This could be explained by incomplete *Oprm1* deletion within the preBötC, or participation of another brain site in OIRD. Injection of opioids into the parabrachial (PBN)/Kolliker-Fuse (KF) nucleus can also slow breathing, making it a candidate second site (*Mustapic et al., 2010*; *Prkic et al., 2012*). In fact, the PBN/KF has been proposed to be the key site mediating OIRD (*Eguchi et al., 1987*; *Lalley et al., 2014*). We therefore took a similar approach to test the role of the PBN/KF in OIRD (*Figure 3A*). AAV-Cre injection into the PBN/KF (*Figure 3—figure supplement 1*) produced a slight increase in the morphine-evoked respiratory rate (*Figure 3—figure supplement 2*, *Figure 3D,E*) and inspiratory airflow (*Figure 3—figure supplement 2*, *Figure 3F,G*), but had a more moderate effect than injection into the preBötC.

To determine if the preBötC and PBN/KF can completely account for OIRD (*Figure 3A*), we genetically deleted *Oprm1* from the preBötC and then from the PBN/KF (Cohort 1) or vice versa (Cohort 2, *Figure 3B*). In either cohort, double deletion breathing after morphine administration looked nearly identical to that of saline control animals (*Figure 3C*), with breathing rate and inspiratory airflow depressed by only ~20% compared to saline (*Figure 3D–G*). Moreover, changes in breathing after viral injection at the second site appeared additive (*Figure 3D–G*) and equivalent to individual preBötC (*Figure 2*) or PBN/KF (*Figure 3—figure supplement 2*, *Figure 3D–G*) effects for each cognate cohort. The double deletion OIRD rescue was similar in normoxia (*Figure 3—figure supplement 3*). To our surprise, rescues also occurred in animals which happened to have mostly unilateral PBN/KF AAV-cre transduction (*Figure 3—figure supplement 4*), therefore these animals were still included in our double deletion analysis (*Figure 3E,G*). Breathing in double-deleted animals was even resilient to super-saturating doses of opioid that severely slow breathing in control animals (150 mg/kg fentanyl, *Figure 3H,I*). Taken together, our data are consistent with a model in which both the preBötC and PBN/KF contribute to opioid respiratory depression, with the former being predominant, and together account for OIRD.

Given the relative importance of the preBötC to OIRD, we sought to identify which *Oprm1* expressing cells within this region depress breathing. Single cell transcriptome profiling of the ventral lateral brainstem of P0 mice (*Figure 4A*) showed that *Oprm1* (mRNA) is expressed almost exclusively by neurons (*Figure 4—figure supplement 1*) and is remarkably restricted to just 8% of presumed preBötC neurons (*Figure 4B*). This alone is interesting, as it suggests that modulation of only a small subset of neurons with the preBötC is enough to significantly impact its ability to generate a rhythm. We also determined that within the preBötC, *Oprm1* (mRNA) was expressed by

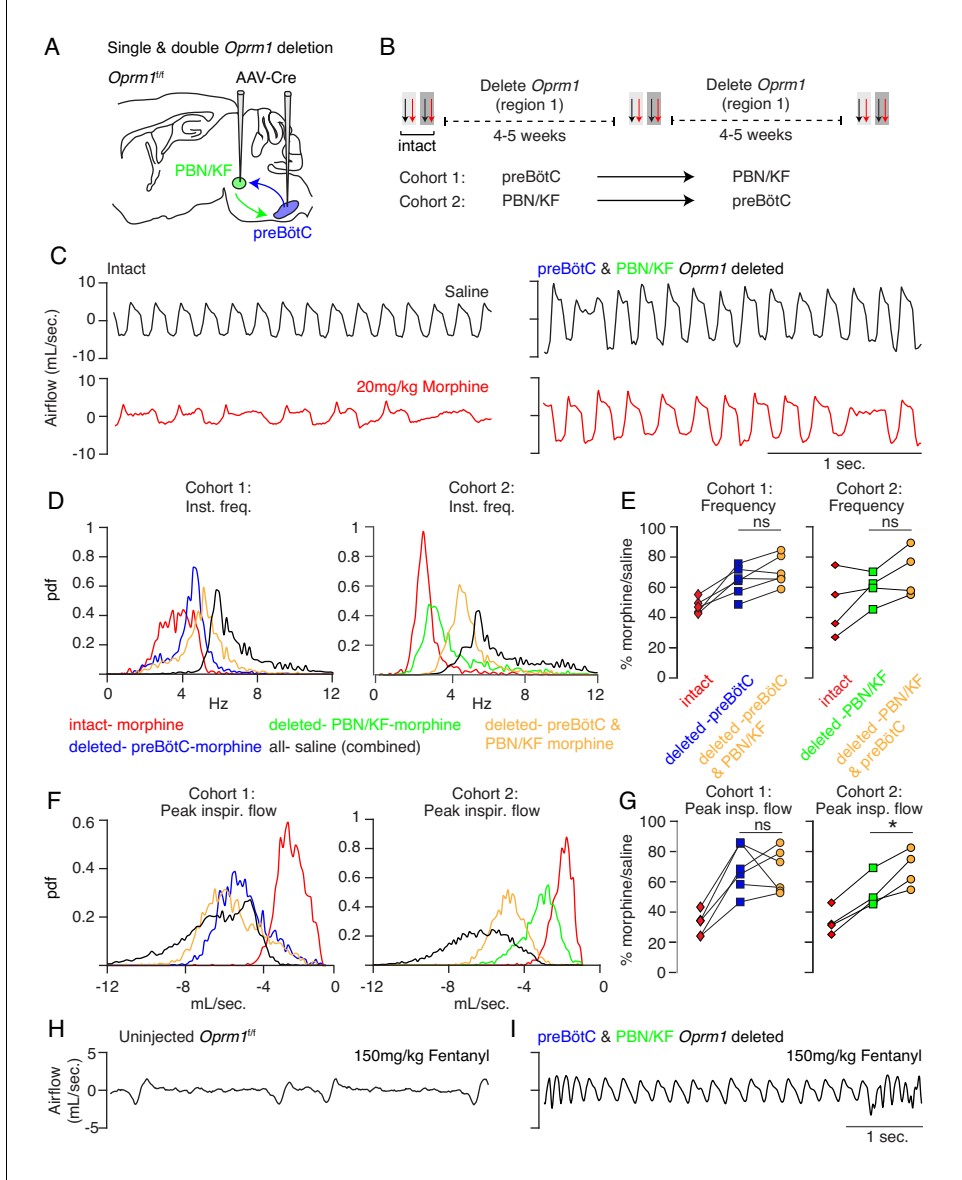

**Figure 3.** Necessity of Parabrachial/Kolliker Fuse nuclei and preBötC in opioid-induced respiratory depression. (**A**) Schematic of a sagittal section through the adult mouse brain showing Cre-viral injection into the preBötC and PBN/KF to measure their individual and combined contribution to OIRD. (**B**) Like *Figure 2*, breathing was assayed before and after each time Cre-virus was injected into *Oprm1^f/f* mice. In cohort 1, Cre-virus was first injected into the preBötC and then the PBN/KF and in cohort 2, Cre-virus was injected into the PBN/KF and then preBötC. (**C**) Representative examples of the breathing airflow (mL/sec.) in hypercapnia for an animal in cohort 1 after saline (black) or morphine (red) before either viral injection and after both preBötC and PBN/KF Cre-virus injections. (**D–E**) Probability density function plot of the instantaneous respiratory frequency for a representative animal from cohorts 1 and 2 (**D**) and ratio of average rate (**E**) after IP injection of morphine-to-saline for 6 animals in cohort 1 and 4 animals in cohort 2 before (pre) and after each Cre-virus injection into the preBötC (blue) or PBN/KF (green). Among the 2 cohorts, n = 6 PBN/KF injections were bilateral and n = 4 mostly unilateral. Cohort 1: preBötC vs. double p-value=0.07, Cohen's d = 0.70. Cohort 2: PBN/KF vs. double p-value=0.1, Cohen's d = 0.76. (**F–G**) Probability density function plot of the peak inspiratory airflow for a representative animal from cohorts 1 and 2 (**F**) and ratio of average peak inspiratory airflow (**G**) after IP injection of morphine-to-saline before (pre) and after each Cre-virus injection into the preBötC (blue) or PBN/KF (green). preBötC Cre-injection has a larger magnitude rescue and after preBötC and PBN/KF injections animals barely have any OIRD phenotype. Cohort 1: preBötC vs. double p-value=0.85, Cohen's d = 0.10. Cohort 2: PBN/KF vs. double p-value=0.03, Cohen's d = 1.32. (**H–I**) Representative plethysmography traces in normoxia from a control *Oprm1^f/f* mouse (**N**) or a double Cre-injected *Oprm1^f/f* mouse

*Figure 3 continued on next page*

*Figure 3 continued*

(preBötC and PBN/KF, (**O**) after IP injection of 150 mg/kg fentanyl. * indicates p-value<0.05. ns indicates p-value>0.05.

The online version of this article includes the following source data and figure supplement(s) for figure 3:

**Source data 1.** Average frequency, peak flow and pause duration after intraperitoneal saline or morphine in hypercapnia at baseline or after *Oprm1* deletion from the preBötC or PBN/KF and then the preBötC and PBN/KF.
**Figure supplement 1.** Expression of GFP protein from AAV-Cre-GFP after bilateral PBN/KF injection.
**Figure supplement 2.** Necessity of Parabrachial/Kolliker Fuse nuclei in opioid-induced respiratory depression.
**Figure supplement 2—source data 1.** Average frequency, peak flow and pause duration after intraperitoneal saline or morphine in hypercapnia at baseline or after *Oprm1* deletion from the PBN/KF.
**Figure supplement 3.** Necessity of the preBötC and PBN/KF for complete opioid induced respiratory depression in normoxia.
**Figure supplement 4.** Expression of GFP protein from AAV-Cre-GFP after mostly unilateral PBN/KF injection.

glycinergic (*Slc6a5* expressing), gabaergic (*Gad2*/*Slc32a1* expressing), and glutamatergic (*Slc17a6* expressing) neural types alike (*Figure 4C*) and therefore *Oprm1* (mRNA) expression is not exclusive to any known rhythmogenic preBötC subpopulation.

Slices containing the preBötC autonomously generate respiratory-like rhythmic activity in vitro which is depressed in both rate and amplitude by bath administration of opioid agonists (*Gray et al., 1999*; *Lorier et al., 2010*), similar to opioid effects we observed on breathing in vivo. To determine which neural class mediates the depression of preBötC activity, we measured rhythmic bursting activity in vitro (*Figure 4D*) after selectively genetically deleting *Oprm1* from each neural class. We achieved this deletion by crossing *Oprm1*^f/f^ mice with each of the following: *Slc17a6*-Cre, *Slc32a1*-Cre, *Gad2*-Cre, or *Slc6a5*-Cre transgenic animals. preBötC slices from control mice (*Oprm1*^f/f^, *Oprm1*^f/+^, or *Oprm1*^+/+^) burst every 5–10 s and this activity was eliminated in 100% of slices by bath application of the selective μ-Opioid receptor agonist [D-Ala$^2$, NMe-Phe$^4$, Gly-ol$^5$]-enkephalin (DAMGO, 50 nM), and subsequently rescued by opioid antagonist naloxone (*Figure 4E,F*). Strikingly, the bursting rhythm of *Slc17a6*-Cre;*Oprm1*^f/f^ slices was not slowed by DAMGO, whereas the rhythm in *Gad2*-, *Slc32a1*-, and *Slc6a5*-Cre; *Oprm1*^f/f^ slices was entirely eliminated, akin to wild type controls (*Figure 4E,F*, *Figure 4—figure supplement 2*). This demonstrates that glutamatergic excitatory neurons, representing ~50% of all preBötC *Oprm1*-expressing neurons and therefore 4% of preBötC neurons, mediate OIRD in vitro.

Next, we dissected the glutamatergic *Oprm1* preBötC neurons by two developmental transcription factors, *Dbx1* or *Foxp2* (*Gray et al., 2010*; *Bouvier et al., 2010*; *Gray, 2013*), to determine if a subset can rescue rhythm depression (*Figure 4G*). Triple-labeling of *Dbx1*-YFP, OPRM1 fused to mCherry (OPRM1-mCherry), and FOXP2 protein quantified within a single preBötC revealed three molecular subtypes of P0 *Oprm1* glutamatergic neurons: 92 ± 9 Dbx1, 50 ± 2 Dbx1/FOXP2, and 20 ± 4 FOXP2 (*Figure 4H,I*). We selectively eliminated the μ-Opioid receptor in these two lineages (*Dbx1*-Cre;*Oprm1*^f/f^ or *Foxp2*-Cre;*Oprm1*^f/f^) and measured preBötC slice activity at increasing concentrations of DAMGO, exceeding the dose necessary to silence the control rhythm (500 nM vs. 50 nM). Elimination of *Oprm1* from both genotypes was sufficient to rescue the frequency and amplitude of preBötC bursting in DAMGO, and the Dbx1 rescue was comparable to elimination of *Oprm1* from all glutamatergic neurons, while the Foxp2 rescue was substantial, but partial (~50–60%, *Figure 4J*). This shows that opioids silence a small cohort (~140) of glutamatergic neurons to depress preBötC activity, and that a molecularly defined subpopulation, about half, can be targeted to rescue these effects.

## Discussion

Here we show that two small brainstem sites are sufficient to rescue opioid induced respiratory depression in vivo. Between them, the preBötC is the critical site and we molecularly define ~140 *Oprm1* glutamatergic neurons within that are responsible for this effect in vitro. Future study of these neurons will provide the first example of endogenous opioid modulation of breathing. Furthermore, characterization of these neurons and their molecular response to opioids may extend existing strategies (*Manzke et al., 2003*) or reveal a novel strategy for separating respiratory depression

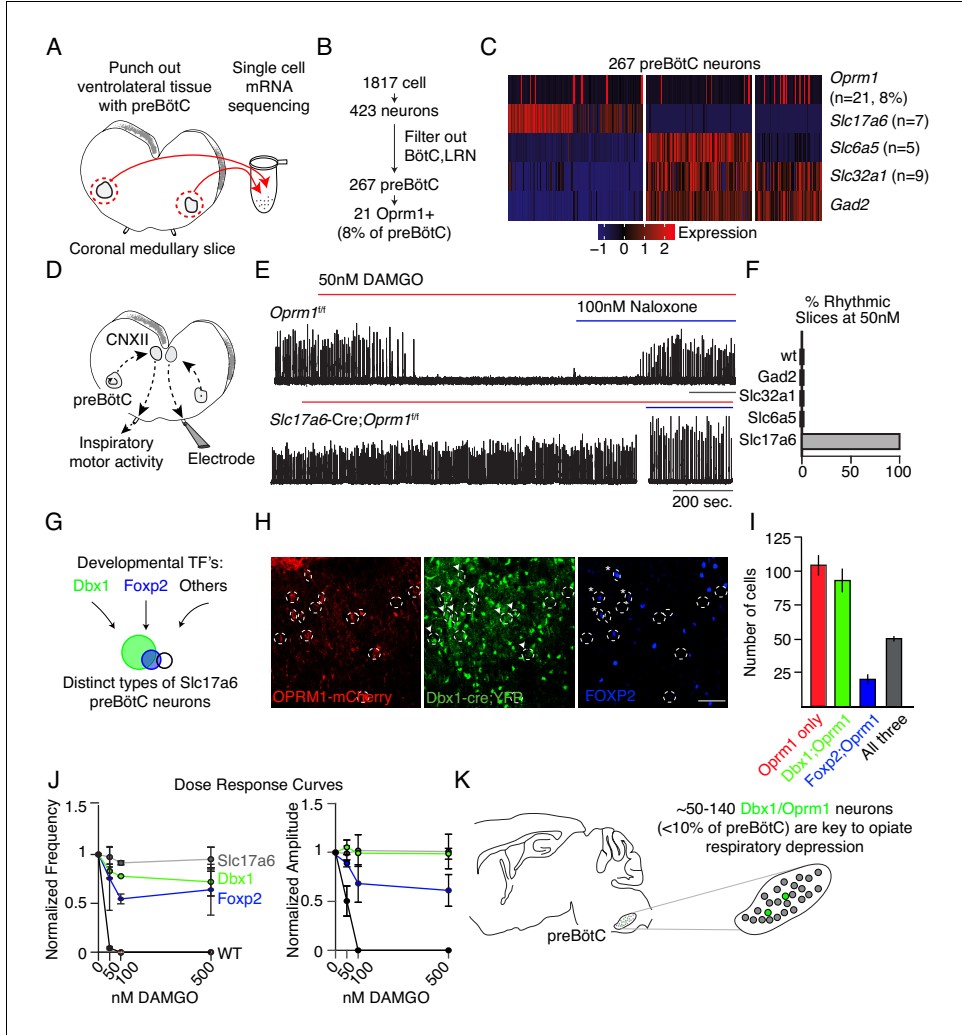

**Figure 4.** Deletion of μ-Opioid receptor from neural subtypes to define their contribution to opioid depression of preBötC burst rhythm and amplitude. (**A**) Schematic of single-cell mRNA sequencing paradigm. Postnatal day 0 (P0) medullary brainstem slices containing the preBötC bilaterally (circled in red) were dissected and isolated for sequencing. (**B**), Single cell transcriptome profiling of cells isolated from P0 preBötC. Of 1817 cells isolated, only 267 were presumed preBötC neurons of which only 21 expressed *Oprm1* mRNA. (**C**), Heatmap of scaled transcript abundance for *Oprm1* and markers of glutamatergic, gabaergic, and glycinergic preBötC neurons. *Oprm1* expressing cells are both excitatory and inhibitory. (**D**), Schematic of extracellular recordings of the preBötC rhythm in P0-4 medullary brainstem slices. The preBötC has input to the hypoglossal motor neurons which form the CN12 rootlet, relaying an inspiratory motor command to the tongue in intact animals. Due to its input from the preBötC, extracellular recording from this rootlet display autonomous rhythmic activity corresponding to in vitro respiration (*Mansour et al., 1994*). (**E**), Representative recording of bursting activity after application of 50 nM DAMGO and 100 nM Naloxone. Top: in control (*Oprm1^{f/f}*) slices bath application of 50 nM DAMGO quickly slowed and decreased the amplitude from baseline bursting. After rhythm cessation, bath application of Naloxone restored the rhythm. Bottom: in *Slc17a6*-cre;*Oprm1^{f/f}* mice 50 nM DAMGO did not stop, or even slow rhythmic activity. (**F**), Percent of slices from each genotype with rhythm cessation after 50 nM DAMGO application. Control (n = 11), *Gad2*-Cre;*Oprm1^{f/f}* (n = 6), *Slc32a1*-Cre;*Oprm1^{f/f}* (n = 2), *Slc6a5*-Cre;*Oprm1^{f/f}* (n = 4), and *Slc17a6*-Cre; *Oprm1^{f/f}* (n = 3). (**G**), Schematic showing three subpopulations of glutamatergic lineages delineated by transcription factors *Dbx1* and *Foxp2*. *Foxp2* neurons represents a smaller, and overlapping, population of *Dbx1* neurons. (**H,**) Identification of molecular subtypes of *Oprm1* preBötC excitatory neurons. Sagittal section of the preBötC from a P0 *Oprm1*-mCherry;*Dbx1*-Cre;Rosa-LSL-YFP mouse immunostained for mCherry (OPRM1 fused to mCherry, red), YFP (Green) and FOXP2 (Blue). About ~50% of *Oprm1* preBötC neurons are glutamatergic/*Dbx1* derived (arrowhead) and of those,~35% express FOXP2 (asterisk). Scale bar, 50 μM. (**I,**) Quantification of the number of preBötC for each molecular subtype identified in **H**. **J**), Dose response curve for the bursting rate and

*Figure 4 continued on next page*

*Figure 4 continued*

amplitude after bath application of 0, 50, 100, and 500 nM DAMGO applied to *Slc17a6*-Cre;*Oprm1*^*f/f*^ (gray, n = 3), *Dbx1*-Cre;*Oprm1*^*f/f*^ (green, n = 3), *Foxp2*-Cre;*Oprm1*^*f/f*^ (blue, n = 4) and control (black, n = 11) P0-4 preBötC slices. Rate and amplitude for each slice are normalized to baseline. (K), Schematic summary showing that the key node for opioids to suppress breathing is the preBötC and within this site, elimination of *Oprm1* from just a small subset of those neurons,~70–140 excitatory neurons, prevents opioid respiratory suppression.

The online version of this article includes the following figure supplement(s) for figure 4:

**Figure supplement 1.** Single cell transcriptome profiling of ventrolateral brainstem neurons.

**Figure supplement 2.** preBötC slice activity in 50 nM DAMGO after deletion of *Oprm1* from inhibitory neural types.

**Figure supplement 3.** Expression of FOXP2 protein and in Foxp2-derived cells within the preBötC.

from analgesia and therefore enable the development of novel opioids or related compounds that relieve pain without risk of overdose.

## A rubric for studying opioid and other respiratory depressants

We find that although multiple breathing parameters are impacted by opioids, decreased inspiratory airflow and delayed breath initiation, which we term pause, represent the primary changes that result in OIRD. Pauses occur during expiration and account for tens to hundreds of milliseconds of low airflow per breath. In hypercapnia, these pause periods terminate with an active expiration. The force, timing of inspiration, and expression of active expiration are ultimately determined by the inspiratory rhythm generator, the preBötC, and focused our initial studies to this site (*Smith et al., 1991*; *Feldman et al., 2013*; *Cui et al., 2016*; *Huckstepp et al., 2016*). Additionally, correlates of these two changes manifest as decreased burst size and frequency in the activity of the preBötC slice in vitro. These two key changes in the breath can guide future OIRD studies and efforts to characterize and test novel opioid drugs. Additionally, this workflow can be applied to the analysis of other respiratory depressants.

## Just two small brainstem sites mediate OIRD

Our experimental design allowed us to determine that both the preBötC and PBN/KF have independent and additive rescue of OIRD. Of these two, the preBötC has the larger magnitude rescue of our two core breathing parameters. The combined deletion of μ-Opioid receptor from both sites essentially eliminates OIRD, even to extremely high doses of the potent opioid fentanyl. This suggests that targeting just these two sites is sufficient to rescue opioid respiratory depression. We interpret the small remaining effect of opioids we observe to be due to incomplete transduction of these brain areas but cannot rule out other minor contributing sites. It is also possible, given the challenge of restricting viral transduction, that some of the demonstrated effects are already due to spillover deletion of μ-Opioid receptor in neighboring brain areas, such as the bulbospinal rostral ventral respiratory group. Our studies were limited to two opioids (morphine and fentanyl) at a specific dose and it will be important in future work to determine if these two brain sites are also critical for OIRD caused by these opioids at different doses or other opioids altogether.

## Depression of preBötC rhythm by silencing a small glutamatergic subpopulation

The two hallmark changes during OIRD, decreased inspiratory airflow and delayed initiation, perfectly match the opioid induced depression of amplitude and frequency in the preBötC slice. We show that ~140 *Oprm1* glutamatergic preBötC neurons mediate this effect. And surprisingly, half this number, just ~70 glutamatergic neurons are sufficient to rescue opioid depression of the preBötC rhythm (50 Dbx1/FOXP2 and 20 FOXP2 in *Foxp2*-Cre;*Oprm1*^*f/f*^, *Figure 4—figure supplement 3*). Further, given the importance of Dbx1 neurons in respiratory rhythm generation (*Gray et al., 2010*; *Bouvier et al., 2010*), rescuing *Oprm1* in just ~50/140 Dbx1 neurons (the Foxp2+ subset) may be sufficient to prevent preBötC depression, the smallest number of neurons we propose. This small number is remarkably consistent with the number of Dbx1 neurons that must be lesioned to arrest preBötC activity (*Wang et al., 2014*). Given the similarity of these effects, we hypothesize that

opioids are primarily acting by silencing presynaptic release, effectively removing these neurons from the network. Alternatively, it is proposed that just a small subset of preBötC excitatory neurons may generate the inspiratory rhythm (*Phillips et al., 2019*) and perhaps these key *Oprm1* neurons are enriched within this subgroup. In this instance, since hyperpolarization of preBötC rhythmogenic neurons slows and silences breathing (*Koizumi et al., 2016*), opioids may act postsynaptically as proposed by others (*Montandon et al., 2011*; *Johnson et al., 1996*). Regardless, it is profound that such a small number can abruptly halt the respiratory rhythm in a network of more than 1000 neurons and suggests that either these neurons act as a key population for rhythmogenesis, or that recurrent excitatory networks are exquisitely sensitive to the number of participating cells. Important future work will need to use the *Dbx1/Oprm1* and *Foxp2/Oprm1* molecular codes to selectively eliminate *Oprm1* from these neurons to test if truly so few neurons profoundly control or modulate breathing in vivo.

# Materials and methods

## Key resources table

| Reagent type (species) or resource | | Source or reference | Identifiers | Additional information |
|---|---|---|---|---|
| Genetic reagent (*Mus musculus*) | *Oprm1*$^{f/f}$ | The Jackson Laboratory | 030074 | RRID:IMSR_JAX:030074 |
| Genetic reagent (*Mus musculus*) | *Oprm1*-mCherry | The Jackson Laboratory | 029013 | RRID:IMSR_JAX:29013 |
| Genetic reagent (*Mus musculus*) | *Slc17a6*-Cre | The Jackson Laboratory | 016963 | RRID:IMSR_JAX:016963 |
| Genetic reagent (*Mus musculus*) | *Gad2*-Cre | The Jackson Laboratory | 0101802 | RRID:IMSR_JAX:0101802 |
| Genetic reagent (*Mus musculus*) | *Slc32a1*-Cre | The Jackson Laboratory | 028862 | RRID:IMSR_JAX:028862 |
| Genetic reagent (*Mus musculus*) | *Slc6a5*-Cre | PMID:25643296 | | |
| Genetic reagent (*Mus musculus*) | *Dbx1*-Cre | PMID:16041369 | | |
| Genetic reagent (*Mus musculus*) | *Foxp2*-Cre | PMID:27210758 | | |
| Genetic reagent (*Mus musculus*) | Rosa-LSL-YFP | The Jackson Laboratory | 006148 | RRID:IMSR_JAX:006148 |
| Adenovirus | AAV5-CMV-Cre-GFP | UNC Vector Core | | AAV5 |
| Adenovirus | AAV5-CAG-GFP | UNC Vector Core | | AAV5 |
| Adenovirus | AAV5-CAG-tdtomato | UNC Vector Core | | AAV5 |
| Antibody | rabbit anti-SST | Peninsula | T-4103 | 1:500 |
| Antibody | rabbit anti-FOXP2 | Abcam | ab16046 | 1:500 |
| Antibody | chicken anti-GFP | Abcam | ab13970 | 1:500 |
| Antibody | rat anti-mCherry | Lifetech | M11217 | 1:500 |
| Antibody | goat anti-rat 555 | Lifetech | A21434 | 1:200 |
| Antibody | goat anti-chicken 488 | Lifetech | A11039 | 1:200 |

*Continued on next page*

*Continued*

| Reagent type (species) or resource | | Source or reference | Identifiers | Additional information |
|---|---|---|---|---|
| Antibody | goat anti-rabbit 633 | Lifetech | 35562 | 1:200 |
| Chemical compound | Morphine sulfate | Henry Schein | 057202 | 20 mg/kg |
| Chemical compound | Fentanyl citrate | Sigma | F3886 | 150 mg/kg |
| Peptide | DAMGO | Abcam | ab120674 | 20–500 nM |

## Animals

*Oprm1$^{f/f}$* (*Weibel et al., 2013*), *Oprm1*-mCherry (*Erbs et al., 2015*), *Slc17a6*-Cre (*Vong et al., 2011*), *Gad2*-Cre (*Taniguchi et al., 2011*), *Slc32a1*-Cre (*Vong et al., 2011*), *Slc6a5*-Cre (*Sherman et al., 2015*), *Dbx1*-Cre (*Bielle et al., 2005*), *Foxp2*-Cre (*Rousso et al., 2016*), Rosa-LSL-YFP (*Madisen et al., 2010*) have been described. Littermates of transgene-containing mice were used as wild type controls. C57Bl/6 mice were used for single cell mRNA sequencing. Mice were housed in a 12 hr light/dark cycle with unrestricted food and water. *Oprm1$^{f/f}$* mice were assigned into experimental and control groups at weaning and given anonymized identities for experimentation and data collection. All animal experiments were performed in accordance with national and institutional guidelines with standard precautions to minimize animal stress and the number of animals used in each experiment.

## Recombinant viruses

All viral procedures followed the Biosafety Guidelines approved by the University of California, San Francisco (UCSF) Institutional Animal Care and Use Program (IACUC) and Institutional Biosafety Committee (IBC). The following viruses were used: AAV5-CMV-Cre-GFP ($4.7 \times 10^{19}$ particles/mL, The Vector Core at the University of North Carolina at Chapel Hill), AAV5-CAG-GFP ($1.0 \times 10^{13}$ particles/mL, The Vector Core at the University of North Carolina at Chapel Hill) or AAV5-CAG-tdtomato ($4.3 \times 10^{12}$ particles/mL, The Vector Core at the University of North Carolina at Chapel Hill).

## Immunostaining

Postnatal day 0–4 brains were dissected in cold PBS, and adult brains were perfused with cold PBS and then 4% paraformaldehyde by intracardiac perfusion. The isolated brains from neonates and adults were then fixed in 4% paraformaldehyde overnight at 4°C and dehydrated in 30% sucrose the next 24 hr at 4°C. Brains were embedded and frozen in OCT once equilibrated in 30% sucrose. Cryosections (18–25 µM) were washed twice for 5 min in 0.1% Tween-20 in PBS, once for 10 min in 0.3% Triton-X100 in PBS, and then twice for 5 min in 0.1% Tween-20 in PBS. Following wash, sections were blocked for 20 min with either 10% goat serum in 0.3% Trition-X100 PBS. Sections were then incubated overnight at 4°C in the appropriate block solution containing primary antibody. Primary antibodies used were: rabbit anti-SST (Peninsula T-4103, 1:500), rabbit anti-FOXP2 (Abcam ab16046, 1:500), chicken anti-GFP (Abcam ab13970, 1:500), rat anti-mCherry (Lifetech. M11217, 1:500). After primary incubation, sections were washed three times for 10 min in 0.1% Tween-20 in PBS, then incubated for 1 hr at room temperature or overnight at 4°C in block containing secondary antibody. Secondary antibodies were: goat anti-rat 555 (Lifetech A21434, 1:200), goat anti-chicken 488 (Lifetech A11039, 1:200), goat anti-rabbit 633 (Lifetech 35562, 1:200). After secondary incubation, sections were washed in 0.1% Tween-20 in PBS and mounted in Mowiol with DAPI mounting media to prevent photobleaching.

## Plethysmography, respiratory and behavioral analysis

Adult (8–20 weeks) *Oprm1$^{f/f}$* mice were first administered either IP 100–200 µL of saline or morphine (20 mg/kg, Henry Schein 057202) and placed in an isolated recovery cage for 15 min to allow full onset of action of the drug. Individual mice were then monitored in a 450 mL whole animal plethysmography chamber at room temperature (22°C) in 21% $O_2$ balanced with $N_2$ (normoxia) or 21% $O_2$,

5% $CO_2$ balanced with $N_2$ (hypercapnia). For fentanyl (150 mg/kg, Sigma F3886) onset of action was so fast (<10 s) that animals were placed directly in the plethysmography chamber after administration of drug. Each session (combination of drug and oxygen condition) was separated by at least 24 hr to allow full recovery. Breathing was monitored by plethysmography, and other activity in the chamber monitored by video recording, for 40 min periods in normoxia and 10 min periods in hypercapnia. In cases where mice were subject to single or double site AAV injection to delete *Oprm1* or sham controls, breathing was recorded first before viral injection and then again after deletion (or sham) more than 4 weeks later. Breathing traces were collected using EMKA iOX2 software and exported to Matlab for analysis. Each breath was automatically segmented based on airflow crossing zero as well as quality control metrics. Respiratory parameters (e.g. peak inspiratory flow, instantaneous frequency, pause length, tidal volume, etc) for each breath, as well as averages across states, were then calculated. Instantaneous frequency was defined as the inverse of breath duration. Pause length was defined as the expiratory period after airflow dropped below 0.5 mL/sec. The pause period is initially a prolonged airflow around or just above 0 mL/sec. and terminates with an increase in expiratory airflow, likely the active expiration induced by hypercapnia (*11*, *Figure 1—figure supplement 1*). The 0.5 mL/sec. threshold was chosen since it identifies low airflow pauses that are just above 0 mL/sec. which rarely occur in control hypercapnic breaths (*Figure 1—figure supplement 2*). Although pauses in length 50-100msec. could be considered false positives because they do not have a considerable low airflow period (see *Figure 1—figure supplement 1 and* , panel 2), they occur at a low rate in saline (1.53%) and we see an increased distribution of pauses in morphine that last hundreds of milliseconds (*Figure 1I*, see *Figure 1—figure supplement 1 and* , panel 3 and 4). Other respiratory parameters were defined by when airflow crosses the value of 0, with positive to negative being inspiration onset and negative to positive being expiration onset. Note, reported airflow in mL/sec. and tidal volume in mL are approximates of the true volumes. Whole body plethysmography imperfectly measures these parameters without corrections for humidity and temperature. However, since humidity and temperature are largely stable between recordings, because they are conducted in a temperature and humidity stable mouse facility, the estimated airflow (mL/sec.) and tidal volume (mL) can be compared in saline vs. morphine or pre and post-Cre virus injection studies. Additionally, in some instances respiratory parameters are appropriately normalized to animal weight in order to accurately compare between animals. However, this normalization is not appropriate for our study since lung volume in mice does not change in adulthood (*Mitzner et al., 2001*), and all respiratory measurements are compared statistically as the ratio of saline to morphine injections within the same animal. The analysis was performed with custom Matlab code available on Github with a sample dataset (*Bachmutsky, 2020*, https://github.com/Yackle-Lab/Opioids-depress-breathing-through-two-small-brainstem-sites; copy archived at https://github.com/elifesciences-publications/Opioids-depress-breathing-through-two-small-brainstem-sites).

Due to limitations in breeding, a power calculation was not explicitly performed before our studies. Studies were conducted on all mice generated; six cohorts of animals. After respiration was measured, mice were sacrificed and injection sites were validated before inclusion of the data for further statistical analysis. We first conducted a Shapiro-Wilk normality on the average values (averaged across breaths) of the pre- and post-morphine respiratory parameters (e.g., peak inspiratory flow, instantaneous frequency) from n = 29 animals. We then used either paired Student's t-test (if normal) or Wilcox Rank Sum test (if not normal) to evaluate statistical significance in comparing the distribution of these values. In comparisons of *Oprm1*-deleted vs. Sham conditions a mixed-repeated measure two-way ANOVA was performed to determine if these two groups were significantly different. Post-hoc Student's t or Wilcox Rank Sum tests were then used to evaluate statistical significance between normalized (morphine/saline, or morphine-saline) respiratory parameters for intact vs. *Oprm1* deleted or intact vs. Sham conditions. Normality in this case was determined by Shapiro-Wilk test on the distribution of normalized respiratory parameters from n = 29 animals. All the above statistics were performed using the publicly available Excel package 'Real Statistics Functions' and SPSS.

## Tail flick assays

Mice were injected with saline (control trials) or 20 mg/kg morphine. 15 min later mice were put into a restraining wire mesh with the tail exposed. One-third of the tail was dipped into a 48–50℃ water bath and time was measured for the tail to flick. Immediately after the flick, the tail was removed

from the bath. If the tail did not flick within 10 s, then the tail was removed. The procedure was video recorded so time to response could be quantified post-hoc. Each mouse was recorded for two saline and two morphine trials.

## Stereotaxic injection

Bilateral stereotaxic injections were performed in mice anesthetized by isoflurane. Coordinates used for the preBötC were: −6.75 mm posterior, −5.05 mm ventral from surface, ±1.3 mm lateral from bregma. Coordinates used for the PBN/KF were: −5.05 mm posterior, −3.7 ventral from surface, ±1.7 lateral from bregma. Injection sites specificity was confirmed by the restricted expression of Cre-GFP, GFP, or tdTomato centered in the anatomically defined Parabrachial/Kolliker-Fuse (*Levitt et al., 2015*) and preBötC (*Smith et al., 1991*; *Feldman et al., 2013*) areas. In the case of preBötC injections, anatomical location of injection site was also confirmed by localization with Somatostatin antibody staining (*Tan et al., 2008*). μ-Opioid receptor deletion was not explicitly demonstrated by immunohistochemistry. After injection of the virus, mice recovered for at least 3–4 weeks before breathing metrics were recorded again. In a subset of animals, mice were then subject to a second site deletion of the complementary brain area, ie. preBötC and then from the PBN/KF (Cohort 1) or vice versa (Cohort 2). These mice were then allowed to recover for another period of at least 3–4 weeks, after which a third set of breathing metrics were recorded. A subset of PBN/KF injected mice had only unilateral expression of Cre and their use is acknowledged in the text.

## Slice electrophysiology

Rhythmic 550 to 650 μm-thick transverse medullary slices which contain the preBötC and cranial nerve XII (XIIn) from neonatal *Oprm1*$^{f/f}$, *Oprm1*$^{f/f}$;*Slc17a6*-Cre+/-, *Oprm1*$^{f/f}$;*Gad2*-Cre+/-, *Oprm1*$^{f/f}$;*Slc6a5*-Cre+/-, *Oprm1*$^{f/f}$;*Slc32a1*-Cre+/-, *Oprm1*$^{f/f}$;*Dbx1*-Cre+/-, *Oprm1*$^{f/f}$;*Foxp2*-Cre+/- (P0-5) were prepared as described (*Ruangkittisakul et al., 2014*). Briefly, slices were cut in ACSF containing (in mM): 124 NaCl, 3 KCl, 1.5 CaCl$_2$, 1 MgSO$_4$, 25 NaHCO$_3$, 0.5 NaH$_2$PO$_4$, and 30 D-glucose, equilibrated with 95% O$_2$ and 5% CO$_2$ (4°C, pH = 7.4). The rostral portion of the slice was taken once the compact nucleus ambiguus was visualized. The dorsal side of each slice containing the closing of the 4$^{th}$ ventricle. For recordings, slices were incubated with ACSF from above and the extracellular K+ was raised to 9 mM and temperature to 27°C. Slices equilibrated for 20 min before experiments were started. The preBötC neural activity was recorded from either XIIn rootlet or as population activity directly from the XII motor nucleus using suction electrodes. Activity was recorded with a MultiClamp700A or B using pClamp9 at 10000 Hz and low/high pass filtered at 3/400 Hz. After equilibration, 20 min. of baseline activity was collected and then increasing concentrations of DAMGO (ab120674) were bath applied (20 nM, 50 nM, 100 nM, 500 nM). Activity was recorded for 20 min. after each DAMGO application. After the rhythm was eliminated or 500 nM DAMGO was reached, 100 nM Naloxone (Sigma Aldrich N7758) was bath applied to demonstrate slice viability. Rhythmic activity was normalized to the first control recording for dose response curves.

## Single cell mRNA sequencing and analysis

650 μm-thick medullary slices containing the preBötC were prepared from 10 P0 mice C57Bl/6 mice as described above. The preBötC and surrounding tissue was punched out of each slice with a P200 pipette tip and incubated in bubbled ACSF containing 1 mg/ml pronase for 30 min at 37°C with intermittent movement. Digested tissue was centrifuged at 800 rpm for 1 min, and the supernatant was discarded and replaced with 1% FBS in bubbled ACSF. The cell suspension was triturated serially with fire-polished pipettes with ~600 μm,~300 μm and ~150 μm diameter. The cells were filtered using a 40 μm cell strainer (Falcon 352340). DAPI was added to a final concentration of 1 μg/mL. The cell suspension was FACS sorted on a BD FACS AriaII for living (DAPI negative) single cells. The cells were centrifuged at 300 g for 5 min and resuspended in 30 μL 0.04% BSA in PBS. The library was prepared using the 10x Genomics Chromium Single Cell 3' Library and Gel Bead Kit v2 (1206267) and according to manufacturer's instructions by the Gladstone genomics core. The final libraries were sequenced on HiSeq 4000.

For analysis, sequencing reads were processed using the 10x Genomics Cell Ranger v.2.01 pipeline. A total of 1860 cells were sequenced. Further analysis was performed using Seurat v2.3. Cells with less than 200 genes were removed from the dataset. Data was LogNormalized and scaled at

1e4. Highly variable genes were identified and used for principal component analysis. 25 principal components were used for unsupervised clustering using the FindCluster function. 12 clusters were identified at a resolution of 1.0, displayed in *Figure 4—figure supplement 1*. FindAllMarkers and violin plots of known cell type markers were used to identify each cluster.

## Acknowledgements

We thank Matthew Collie (currently Harvard University) for his scientific input throughout the project and experimental contributions to the slice electrophysiology. We thank Drs. David Julius and Roger Nicoll (University of California, San Francisco) for input and revision of the manuscript. We thank Dr. Massimo Scanziani (University of California, San Francisco) for mentorship, discussion, and revision of the manuscript. We thank Adelae Durand (Yackle lab, University of California, San Francisco) for data collection and her revision of the manuscript.

## Additional information

### Funding

| Funder | Grant reference number | Author |
|---|---|---|
| NIH Office of the Director | DP5-OD023116 | Kevin Yackle |
| University of California, San Francisco | Program for Breakthrough Biomedical Research | Kevin Yackle |

The funders had no role in study design, data collection and interpretation, or the decision to submit the work for publication.

### Author contributions

Iris Bachmutsky, Conceptualization, Data curation, Software, Formal analysis, Validation, Investigation, Visualization, Methodology, Writing - original draft, Writing - review and editing; Xin Paul Wei, Data curation, Formal analysis, Investigation, Methodology; Eszter Kish, Conceptualization, Investigation, Methodology; Kevin Yackle, Conceptualization, Resources, Supervision, Funding acquisition, Investigation, Visualization, Methodology, Writing - original draft, Project administration, Writing - review and editing

### Author ORCIDs

Kevin Yackle (iD) https://orcid.org/0000-0003-1870-2759

### Decision letter and Author response

Decision letter https://doi.org/10.7554/eLife.52694.sa1
Author response https://doi.org/10.7554/eLife.52694.sa2

## Additional files

### Supplementary files

• Supplementary file 1. Animal behavior during recordings of breathing in normoxia and hypercapnia.
• Transparent reporting form

### Data availability

Summary data generated in this study are included as a supplemental supporting file. All Matlab code and an example data are posted on Github: https://github.com/YackleLab/Opioids-depress-breathing-through-two-small-brainstem-sites (copy archived at https://github.com/elifesciences-publications/Opioids-depress-breathing-through-two-small-brainstem-sites).

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
