## [Decision Letter]

**Acceptance summary:**

This is a technically sophisticated study employing mouse transgenic approaches, viral vector-based neuronal targeting, in vivo behavioral analyses, single cell transcriptomics, and in vitro electrophysiological measurements to explore the mechanisms underlying opioid respiratory depression. The tour-de-force of impressive tools and the clear presentation make this paper especially appealing to a wide audience beyond those who study respiration.

The authors determine that mu-opioid receptors (Oprm1) in neurons in two circumscribed brainstem sites- the preBötzinger Complex (preBötC) and parabrachial (PBN)/Kolliker-Fuse (KF) nucleus, account for opioid-induced respiratory depression (OIRD) in awake behaving adult mice in vivo as well as in rhythmically-active preBötC circuits in slices from neonatal mice in vitro. Importantly between the preBötC and PBN/KF, the preBötC is the more critical site in vivo, and that at least in the slices in vitro, a small subpopulation (~140) Oprm1-expressing glutamatergic neurons, including Dbx1and Foxp2 expressing glutamatergic neurons, within the preBötC are responsible for the opioid suppression of rhythmic activity.

**Decision letter after peer review:**

Thank you for submitting your article "Opioids depress breathing through two small brainstem sites" for consideration by *eLife*. Your article has been reviewed by three peer reviewers, including Jan-Marino Ramirez as the Reviewing Editor and Reviewer #1, and the evaluation has been overseen by Eve Marder as the Senior Editor. The following individuals involved in review of your submission have agreed to reveal their identity: Gaspard Montandon (Reviewer #2); Jeffrey C. Smith (Reviewer #3).

The reviewers have discussed the reviews with one another and the Reviewing Editor has drafted this decision to help you prepare a revised submission.

Summary:

This is a technically sophisticated study employing mouse transgenic approaches, viral vector-based neuronal targeting, in vivo behavioral analyses, single cell transcriptomics, and in vitro electrophysiological measurements demonstrating that mu-opioid receptors (Oprm1) in neurons in two circumscribed brainstem sites- the preBötC and PBN/KF, account for opioid-induced respiratory depression (OIRD) in awake behaving adult mice in vivo as well as in rhythmically-active preBötC circuits in slices from neonatal mice in vitro. The experimental approaches employed are novel, providing a number of important results that more clearly define the critical brainstem sites and molecular phenotypes of neurons involved in OIRD. The authors' strategy of Cre-viral injection into the preBötC and/or PBN/KF of Oprm1 (f/f) mice to genetically delete Oprm1 together with the in vivo behavioral analyses to measure the individual and combined contribution of these two functionally defined regions to OIRD proved to be very effective. Also very effective is their use of Vglut2-Cre, Vgat-Cre, Gad2-Cre, or Glyt2-Cre transgenic mouse lines crossed with the Oprm1 (f/f) mice to selectively genetically delete Oprm1 from different neuronal classes, as well as the dissection of the glutamatergic Oprm1 preBötC neurons by two developmental transcription factors, Dbx1 or Foxp2, all combined with electrophysiological analyses of the rhythmic activity responses to opioid receptor agonists in slices in vitro. The important results are that between the preBötC and PBN/KF, the preBötC is the more critical site in vivo, and that at least in the slices in vitro, a small subpopulation (~140) Oprm1-expressing glutamatergic neurons, including Dbx1and Foxp2 expressing glutamatergic neurons, within the preBötC are responsible for the opioid suppression of rhythmic activity. The paper is well written and the figures and supplementary material very nicely illustrate the essential experimental results. However, there are several concerns that need to be addressed.

Essential revisions:

1) The authors try to introduce a "gold standard" definition for how breathing changes in OIRD. Yes, it would be nice if there were a simple gold standard. However, this is not the reality. Gold standard for OIRD should come from a consensus of studies showing similar results. The research presented here show one example of respiratory depression with only one type of opioids, not highly specific for the µ-opioid receptors, at only one dosage. Also the "gold standard" proposed here is based upon flow which cannot be reliably measured with whole-body plethysmography. So don't state that you are introducing a gold standard. OIRD is very state dependent and shows huge interindividual variability. Trying to imply that the respiratory rhythm is simply depressed is an oversimplification. Just look at the huge variability in Te (Figure 1H). Plethysmograph recordings are extremely difficult to evaluate, because animals behave, morphine will be sedating before measuring the OIRD, this will not be a static response. We don't understand the definition of inspiratory time, expiratory time, and pause. It seems in Figure 1G that flow doesn't cross the zero at the end of expiration as shown in the figure, but rather at the end of the pause. Is the "pause" only a longer expiration? How was this pause detected or defined? If those pauses are part of expiration, then expiratory time is also increase by morphine. How was the criterion for pause of 0.5mL/sec defined? Knowing that flow is highly variable from one animal to the other it is very likely an unreliable threshold.

Along the same lines: The authors respiratory assessments in freely-behaving rodents limited for a few reasons: a) using an arbitrary threshold based on flow to define the end of expiration is not reliable. Flow can change from one animal to another, so the duration of their "pause" may differ depending upon the initial baseline of the animal. How do you determine the amplitude, where do you set the threshold = look at Figure 1A, what do you consider a breath and what not?

b) Using tidal volume or flow as measurements in whole-body plethysmography in rodents is problematic. Unless the right formula is used, it is very difficult to evaluate tidal volume in rodents. In addition, mL cannot be a correct measurement in rodents without normalizing the value according to body weight. Unless all animals have the exact body weight, flow and tidal volume need to be measured as ml/10g or other units. You need to carefully consider these caveats

c) One of the difficult of measuring breathing with plethysmography is to control for arousal levels. It is well know that sedation by opioids may affect the severity of respiratory depression so arousal levels before morphine may be an issue. Please discuss this important caveat.

2) While using hypercapnia as a tool may simplify the approach, it will dramatically change the behavioral state of the animal. Under hypercapnia, arousal levels may be severely disrupted and it has been shown that arousal levels are tightly linked to respiratory depression. Hypercapnia will also affect the opiate sensitivity, and this effect will not be uniform in different areas. Carefully discuss this important caveat, and don't imply that this is an advantage, other than the fact that it regularizes the rhythm.

3) Figure 2 characterizes the necessity of PreBötC. Without quantitative immunohistochemical confirmation (Figure 2—figure supplement 1) it is indeed difficult to know whether the injections were restricted to the preBötC, same applies to the KF injections. (Figure 3—figure supplement 1).

4) Oprm1-/- animals were not compared with other mice that received injection of scrambled AAV, but were compared to themselves before surgery, microinjection, and recovery. It is possible that the inflammation induced by AAV, and the damage done by the microinjection may change the response to opioids. We recommend that this should be verified with proper controls.

5) From the transcriptome profiling and neonatal in vitro results, the authors propose that opioids silence a small subset (~140) of glutamatergic neurons to depress preBötC activity, and that a molecularly defined subpopulation (~70 neurons) can be targeted to rescue these effects. It would be important to emphasize in the Discussion that tests need to be performed in the awake behaving mice in vivo with the transgenic lines utilized for the in vitro studies to determine if so few neurons are also responsible for OIRD in the adult system in vivo.

[Editors' note: further revisions were suggested prior to acceptance, as described below.]

Thank you for resubmitting your article "Opioids depress breathing through two small brainstem sites" for consideration by *eLife*. Your article has been reviewed by two peer reviewers, and the evaluation has been overseen by Ronald Calabrese as the Senior Editor and Reviewing Editor. The following individuals involved in review of your submission have agreed to reveal their identity: Gaspard Montandon (Reviewer #2); Jeffrey C Smith (Reviewer #3).

The reviewers have discussed the reviews with one another and the Reviewing Editor has drafted this decision to help you prepare a revised submission.

Summary:

This paper has been previously reviewed and requires some further revisions. All these revisions can be accomplished without any further experiments. Some new analysis in needed but the vast majority of the changes involve rewriting.

Essential revisions:

There are still some outstanding issues that must be addressed.

Anatomy a) Regarding PBN/KF injection sites, it is clear that these encompassed a much larger area than the targeted region, which is not surprising for targeting these small areas in the mouse brainstem, although it looks like (at least from the examples shown) that the most concentrated fluorescent protein expression may typically be in and immediately around the targeted areas. So it is clear that the authors have included the targeted areas with their stereotaxic injection approach.

b) Regarding preBötC targeting, the reconstructions shown in Figure 2—figure supplement 1, part A and the two anterior sites in part B are reasonable for demonstrating preBötC targeting in the mouse, but the labeling for the posterior site reconstructed extends too caudal as the authors have represented the anatomy, implying some receptor deletion in the rVRG. The authors thusly should modify statements in the manuscript regarding site-specificity and that injection sites were confirmed by the restricted expression of Cre-GFP (subsection “Stereotaxic injection”, and figure legends), and they need to discuss any caveats associated with this issue.

c) Given the results showing functional rescue from depressed breathing frequency and peak inspiratory flow obtained with their Oprm1f/f transgenic and Cre vector construct, their results with control vector injections, and the injection site anatomical reconstructions, it is likely that the authors have deleted some opioid receptors in the regions of interest. We ask the authors to further clarify in the manuscript if/how they have validated deletion of opioid receptors with their experimental approach and to address any associated caveats. The authors could confirm the absence of MOR in knockout animals with Immunohistochemistry.

Hypercapnia

Measuring breathing in 5% CO2 likely limits the authors' conclusions. They need to mention that these data are valid in animals that are exposed to CO2, and discuss the issues related to high CO2, low pH, hyperventilation induced by CO2, and reduced arousal, which may limit the role of the preBotC in opioid-induced respiratory depression. The authors don't mention any papers in the discussion related to hypercapnia, medullary sites sensitive to CO2, arousal states, opioid receptor knockout animals etc.

Pauses

Definition of pause versus expiration. The definition of a pause according to the authors is when volume goes below 0.5mL during the expiratory period. According to Figure 1G, the last part of the pause with morphine seems to be above this threshold. This arbitrary threshold is concerning. The threshold is likely detecting different pauses in saline versus morphine considering that volume is considerably reduced by morphine. For instance, in Figure 1G, peak flow with saline condition is -6 mL/sec versus -2ml/sec for morphine, with corresponding differences in volume. The use of an absolute value (0.5mL) as a threshold means that the detection of the "pause" will likely be different between saline and morphine conditions, so not due to different timings but different detections. It would be easier to use the simple definitions of inspiratory and expiratory times, which would be consistent with previous reports showing that expiratory duration is affected by opioids. Also, we are not sure what the physiological relevance of a pause is. Is it controlled by different brainstem circuits? The long expirations are likely due to a combination of hypercapnia and hypoxia with morphine.

Discussion

The authors do not discuss previous studies about MOR-/-, preBötC, etc. There are many studies investigating this topic that could be discussed in the Discussion. Dahan et al., 2001, Montandon et al., 2011, 2016, Varga et al., 2020, Stucke et al., Mustapic et al., Lalley et al. etc.

---

## [Author Response]

Essential revisions:1) The authors try to introduce a "gold standard" definition for how breathing changes in OIRD. Yes, it would be nice if there were a simple gold standard. However, this is not the reality. Gold standard for OIRD should come from a consensus of studies showing similar results. The research presented here show one example of respiratory depression with only one type of opioids, not highly specific for the µ-opioid receptors, at only one dosage. Also the "gold standard" proposed here is based upon flow which cannot be reliably measured with whole-body plethysmography. So don't state that you are introducing a gold standard. OIRD is very state dependent and shows huge interindividual variability. Trying to imply that the respiratory rhythm is simply depressed is an oversimplification. Just look at the huge variability in Te (Figure 1H). Plethysmograph recordings are extremely difficult to evaluate, because animals behave, morphine will be sedating before measuring the OIRD, this will not be a static response.We don't understand the definition of inspiratory time, expiratory time, and pause.

1) Inspiratory time (Ti) is defined as the period of negative airflow at the beginning of the breath, such that airflow is less than 0mL/sec. Once the 0mL/sec. threshold is crossed to positive values, inspiration is considered to be terminated.

2) Expiratory time (Te) is defined as the period of positive airflow immediately after inspiration, when the 0mL/sec. threshold is crossed until the airflow decreases below a second threshold of 0.5mL/sec. We use this second threshold to distinguish Te from the pause. We have now clarified in the manuscript that this is the initial portion of expiration (Results, second paragraph, subsection “A rubric for studying opioid and other respiratory depressants”, subsection “Plethysmography, respiratory and behavioral analysis”, first paragraph).

3) The pause is defined as the airflow after Te until the onset of the next inspiration. The specific details of the pause are discussed below.

It seems in Figure 1G that flow doesn't cross the zero at the end of expiration as shown in the figure, but rather at the end of the pause. Is the "pause" only a longer expiration?

The pause is an expiratory period that occurs after the initial large-amplitude expiratory pulse wherein the majority of tidal volume has been expelled. The pause is characterized by very low airflow and ends with a final brief expulsion of air. Since this phenomenon is being described in hypercapnia, we interpret this terminating expiratory airflow to be "active expiration" (Pisanski and Pagliardini, 2019). The low airflow pause period can vary in length from tens to hundreds of milliseconds. We have now clarified this definition in the second paragraph of the Results, subsection “A rubric for studying opioid and other respiratory depressants” and in the first paragraph of the subsection “Plethysmography, respiratory and behavioral analysis” and have added a supplementary figure (Figure 1—figure supplement 1) which shows breaths with different pause lengths aligned by the onset of the pause to better qualitatively demonstrate this phenomenon. Additionally, we have added a description in the Materials and methods and Figure 1—figure supplement 2 to clarify how the airflow threshold was chosen to define pause onset.

How was this pause detected or defined?

The pause is an apneic period of low airflow which we believe represents a failure or delay to initiate a subsequent breath. This period is qualitatively defined as the point during expiration when expiratory airflow crosses below 0.5mL/sec. All airflow after this point, until the next inspiration begins (continuous airflow below 0mL/sec.) is defined as the pause. In control hypercapnic breathing fewer than 2% of breaths have a pause longer than 5msec. by this definition (Figure 1—figure supplement 2). After morphine, ~40% of breaths have a pause between 5 and several hundred milliseconds (Figure 1—figure supplement 2). This definition is now clarified in the first paragraph of the subsection “Plethysmography, respiratory and behavioral analysis”and we have added a supplementary figure (Figure 1—figure supplement 2).

If those pauses are part of expiration, then expiratory time is also increase by morphine.

It is certainly true that the overall time between two inspirations does increases. However, the initial portion of expiration (before our pause) does not change in length. This is the data displayed in Figure 1H. We have now clarified our language to include the pause as part of the overall expiratory period in the second paragraph of the Results, subsection “A rubric for studying opioid and other respiratory depressants” and in the first paragraph of the subsection “Plethysmography, respiratory and behavioral analysis”.

How was the criterion for pause of 0.5mL/sec defined?

Please see discussion above and Figure 1—figure supplement 2.

Knowing that flow is highly variable from one animal to the other it is very likely an unreliable threshold.

Since the pause analysis was conducted on data from adult animals in hypercapnia, airflow recordings are very reproducible between animals and not "highly variable". For example, for 29 animals recorded, the mean+/-SD for average peak inspiratory airflow after saline injection was 7.2+/-0.57mL/sec. and after morphine injection was 3.60+/-0.62mL/sec. Furthermore, in Figure 1—figure supplement 2, where the pause threshold is defined, the data in the scatter plot was generated from 100 breaths from 15 different animals and all data points are intermixed and not clustering as individual animals. Therefore, our pause threshold is chosen to be a reliable metric between animals.

Along the same lines: The authors respiratory assessments in freely-behaving rodents limited for a few reasons: a) Using an arbitrary threshold based on flow to define the end of expiration is not reliable. Flow can change from one animal to another, so the duration of their "pause" may differ depending upon the initial baseline of the animal. How do you determine the amplitude, where do you set the threshold = look at Figure 1A, what do you consider a breath and what not?

Please see explanations above and Figure 1—figure supplement 2. Briefly, data and pause threshold between animals is highly reproducible and data from 29 animals was used to define the pause threshold.

b) Using tidal volume or flow as measurements in whole-body plethysmography in rodents is problematic. Unless the right formula is used, it is very difficult to evaluate tidal volume in rodents.

We appreciate the criticism that whole-body plethysmography inaccurately measures tidal volume and that a more correct estimate necessitates the incorporation of chamber humidity and temperature via the Drorbaugh and Fenn formula. We have now indicated that the displayed tidal volumes are approximates in the first and third paragraphs of the Results, the first paragraph of the subsection “Plethysmography, respiratory and behavioral analysis”, and in the legend of Figure 1C.

In addition, mL cannot be a correct measurement in rodents without normalizing the value according to body weight. Unless all animals have the exact body weight, flow and tidal volume need to be measured as ml/10g or other units. You need to carefully consider these caveats.

It is certainly important to normalize the airflow and tidal volume to body weight when animals of very different sizes are being compared. For example, this correction is particularly important when comparing developing rodents where lung size strongly correlates with size. However, we feel that this correction is not appropriate in our manuscript for four reasons:

1) When different animals are being compared, the respiratory parameters after morphine injection are normalized to those after saline injection recordings. These two recordings are performed within several days of each other (i.e. without significant weight change). All statistical tests are performed on normalized data.

2) As mentioned above, one mechanism that could explain changes in airflow in differently sized mice is if lung volume increased with body size. However once mice reach their adult size, although their body weight can increase, their lung volume does not. This has been clearly demonstrated with serial computed tomography imaging of lung size (Mitzner, Brown and Lee, 2001).

3) Another possible mechanism that could contribute to changes in airflow across different sized mice is if increased displacement of air in the plethysmography chamber distorts the airflow signal. However, in a toy experiment wherein the chamber is filled with objects that displace either 20, 25, or 30mL’s of air, the total volume of air injected by syringe (10ml) can be accurately calculated in all conditions. (See Author response image 1). This indicates that normalizing approximate airflow or tidal volume to weight, as a proxy for body volume, will negatively impact the accuracy of those estimates.

4) We can demonstrate the stability of airflow across recordings over time, even though the animal has gained some weight. When airflows from the same animal, after injection of saline, are compared at the beginning and end of our studies, the airflows are indistinguishable despite some mice increasing in weight (See Author response image 2 where specific breaths in hypercapnia are compared). In this case normalizing by weight would lead the experimenter to believe lung volume significantly decreases as animals increase in weight which would have no biological basis.

In summary, it is unlikely that either animal weight or size in adult mice is significantly biasing approximated respiratory airflow or tidal volume values, and therefore it would be inappropriate to normalize these values by weight.

**Author response image 1. respfig1:** The plethysmograph was filled with either a 20, 25 or 30mL object to approximate a 20, 25, or 30 gram mouse. The plethysmograph was injected with 10mL of air and the change in pressure was integrated to calculate a tidal volume. The calculated tidal volumes were all centered at 10mL’s and the size of the object did not impact the calculated volume.

**Author response image 2. respfig2:** Average +/- SD (top) and example breaths (bottom) that have Ti between 50-100msec. from the same animal during control hypercapnia recordings from the beginning of our protocol (left) versus the end (right). Notice that although the mouse has gained weight, the airflow of these breaths has not changed.

c) One of the difficult of measuring breathing with plethysmography is to control for arousal levels. It is well know that sedation by opioids may affect the severity of respiratory depression so arousal levels before morphine may be an issue. Please discuss this important caveat.

We certainly appreciate that arousal levels impact the distribution of breaths that are taken by awake, behaving mice and in fact, this concern is exactly why we decided to measure breathing before and after morphine in hypercapnia in addition to normoxia (see Results, first paragraph). Although hypercapnia stimulates the respiratory system, it also normalizes the behavior and we presume the arousal state of mice. As a result, as we show in Figure 1, all of the changes in respiratory parameters are more reproducible between mice.

2) While using hypercapnia as a tool may simplify the approach, it will dramatically change the behavioral state of the animal. Under hypercapnia, arousal levels may be severely disrupted and it has been shown that arousal levels are tightly linked to respiratory depression. Hypercapnia will also affect the opiate sensitivity, and this effect will not be uniform in different areas. Carefully discuss this important caveat, and don't imply that this is an advantage, other than the fact that it regularizes the rhythm.

Although it is certain that hypercapnia alters arousal levels, breathing, and sensitivity to opioids, we use the regularization of breathing as a way to dissect the role of the preBötC and PBN/KF in OIRD. We acknowledge the benefit of hypercapnia (Results, first paragraph) is simply that it regularizes the respiratory rhythm which is in large part due to a reduction of behaviors performed, for example, far less sniffing and grooming. Although the changes to breathing in hypercapnia are quantitatively distinct from normoxia (Figure 1), the ability to eliminate the variability in breathing due to behavior enabled us to quantitively measure how breaths change in morphine and how these changes can be rescued after deletion of Oprm1 in the preBötC and/or PBN/KF sites.

Even if hypercapnia produces an altered arousal state that changes the OIRD response, we do not anticipate that it is due to the "masking" a third brain site that it critical for ORID in normoxia. This is because in all the animals studied, we also performed recordings in normoxia that demonstrate a rescue in respiratory rate and peak inspiratory flow after deletion of Oprm1 from the preBötC and then the PBN/KF which we now include in additional supplementary figures (Figure 2—figure supplement 2, Figure 3—figure supplement 3). This demonstrates that the rescues we observe are not unique to hypercapnia.

3) Figure 2 characterizes the necessity of PreBötC. Without quantitative immunohistochemical confirmation (Figure 2—figure supplement 1) it is indeed difficult to know whether the injections were restricted to the preBötC, same applies to the KF injections. (Figure 3—figure supplement 1).

We have now modified the supplementary figures that detail the preBötC and PBN/KF viral injections (Figure 2—figure supplement 1 and Figure 3—figure supplement 1). These figures now contain a schematic that details the brain anatomy from anterior to posterior brain sections and the extent of Cre-GFP expression from each animal. Note that the overlapping GFP-expressing region from all animals is the preBötC and PBN/KF.

4) Oprm1-/- animals were not compared with other mice that received injection of scrambled AAV, but were compared to themselves before surgery, microinjection, and recovery. It is possible that the inflammation induced by AAV, and the damage done by the microinjection may change the response to opioids. We recommend that this should be verified with proper controls.

We have now used a mixed repeated measures two-way ANOVA to compare our preBötC AAV-Cre injected animals with the control preBötC AAV-GFP or AAV-TdTomato injected animals (Figure 2 legend). Additionally, we have added 3 new control preBötC injected animals (new total is n=7) in order to match or exceed number of experimental animals (n=6). The addition of new mice has verified our original observations of no change in frequency, airflow and pause in control injected animals.

5) From the transcriptome profiling and neonatal in vitro results, the authors propose that opioids silence a small subset (~140) of glutamatergic neurons to depress preBötC activity, and that a molecularly defined subpopulation (~70 neurons) can be targeted to rescue these effects. It would be important to emphasize in the Discussion that tests need to be performed in the awake behaving mice in vivo with the transgenic lines utilized for the in vitro studies to determine if so few neurons are also responsible for OIRD in the adult system in vivo.

The limited expression of Oprm1 within the preBötC coupled with the in vivo OIRD rescue observed after deletion of Oprm1 from the preBötC makes (which is likely even incomplete, i.e. not all cells are expressing viral), makes the powerful prediction that just a small number of neurons is capable of dramatically changing the respiratory rhythm in vivo. Still, at the end of the Discussion we have now clarified that this is an important hypothesis that future experiment should test directly.

[Editors' note: further revisions were suggested prior to acceptance, as described below.]

Please note, during the course of the revision we validated that our dose of morphine (20mg/kg) does not change respiration in mice completely lacking µOR-/-. This demonstrates that this dose of morphine is specific to µOR. As a result, we have removed a phrase in our Discussion suggesting this dose of morphine could activate kappa and δ opioid receptors (per the request of the reviewer in revision round 1).

Essential revisions:There are still some outstanding issues that must be addressed.Anatomy a) Regarding PBN/KF injection sites, it is clear that these encompassed a much larger area than the targeted region, which is not surprising for targeting these small areas in the mouse brainstem, although it looks like (at least from the examples shown) that the most concentrated fluorescent protein expression may typically be in and immediately around the targeted areas. So it is clear that the authors have included the targeted areas with their stereotaxic injection approach.

Ok.

b) Regarding preBötC targeting, the reconstructions shown in Figure 2—figure supplement 1, part A and the two anterior sites in part B are reasonable for demonstrating preBötC targeting in the mouse, but the labeling for the posterior site reconstructed extends too caudal as the authors have represented the anatomy, implying some receptor deletion in the rVRG. The authors thusly should modify statements in the manuscript regarding site-specificity and that injection sites were confirmed by the restricted expression of Cre-GFP (subsection “Stereotaxic injection”, and figure legends), and they need to discuss any caveats associated with this issue.

Subsection “Stereotaxic injection” now indicates that the site-specificity is indicated by the center of the Cre-GFP expression. We also have added additional discussion (subsection “Just two small brainstem sites mediate OIRD”) to note that we cannot exclude µOR deletion from neighboring regions.

c) Given the results showing functional rescue from depressed breathing frequency and peak inspiratory flow obtained with their Oprm1f/f transgenic and Cre vector construct, their results with control vector injections, and the injection site anatomical reconstructions, it is likely that the authors have deleted some opioid receptors in the regions of interest. We ask the authors to further clarify in the manuscript if/how they have validated deletion of opioid receptors with their experimental approach and to address any associated caveats. The authors could confirm the absence of MOR in knockout animals with Immunohistochemistry.

We clarified our methodology and describe that the Cre-GFP determined injection site-specificity indicates where the µ-opioid receptor deleted. Although we infer a deletion by the rescue of OIRD, we indicate that we do not explicitly confirm deletion by immunohistochemistry (Results, fourth paragraph and subsection “Stereotaxic injection”).

HypercapniaMeasuring breathing in 5% CO2 likely limits the authors' conclusions. They need to mention that these data are valid in animals that are exposed to CO2, and discuss the issues related to high CO2, low pH, hyperventilation induced by CO2, and reduced arousal, which may limit the role of the preBotC in opioid-induced respiratory depression. The authors don't mention any papers in the discussion related to hypercapnia, medullary sites sensitive to CO2, arousal states, opioid receptor knockout animals etc.

We disagree with the reviewer that our conclusions are limited to 5% CO2 and a limited role of the preBötC for two simple reasons:

1) Much of the presented data are in under 5% CO2 conditions for uniformity and clarity of data presentation. However, as indicated in our experimental methods and Figure 2—figure supplement 2 and Figure 3—figure supplement 3, the OIRD rescues in peak inspiratory flow and frequency observed in 5% CO2 also occur in normoxia.

2) If 5% CO2 “[limits] the role of the preBötC in opioid induced respiratory depression” (as suggested above), then elimination of µOR from the preBötC would not result in rescued OIRD. This is simply not what we observe.

PausesDefinition of pause versus expiration. The definition of a pause according to the authors is when volume goes below 0.5mL during the expiratory period. According to Figure 1G, the last part of the pause with morphine seems to be above this threshold. This arbitrary threshold is concerning. The threshold is likely detecting different pauses in saline versus morphine considering that volume is considerably reduced by morphine. For instance, in Figure 1G, peak flow with saline condition is -6 mL/sec versus -2ml/sec for morphine, with corresponding differences in volume. The use of an absolute value (0.5mL) as a threshold means that the detection of the "pause" will likely be different between saline and morphine conditions, so not due to different timings but different detections. It would be easier to use the simple definitions of inspiratory and expiratory times, which would be consistent with previous reports showing that expiratory duration is affected by opioids. Also, we are not sure what the physiological relevance of a pause is. Is it controlled by different brainstem circuits? The long expirations are likely due to a combination of hypercapnia and hypoxia with morphine.

We believe that by introducing the concept of a “pause” period is important for two reasons: 1) it is the most accurate way to describe our data and 2) reflective of the underlying physiology.

1) Perhaps the most obvious qualitative change in breathing under morphine are that many breaths have long periods (lasting several hundred milliseconds at 20mg/kg morphine) of little to no airflow (Figure 1A and D, Figure 1—figure supplement 1). We sought to rigorously and quantitatively describe this phenomenon. As we show in Figure 1—figure supplement 2, we do not “arbitrarily” choose the 0.5mL/sec. threshold, and instead chose this threshold because it identifies essentially no pauses in hypercapnia after saline injection (only 1.53% of breaths have a pause). Although pauses in length 50-100msec. could be considered false positives because they do not have a considerable apneic period (see Figure 1—figure supplement 1, panel 2), they occur at a low rate in saline (1.53%) and we see an increased distribution of pauses in morphine that last hundreds of milliseconds (Figure 1I, see Figure 1—figure supplement 1, panel 3 and 4). We now clarify this in the first paragraph of the subsection “Plethysmography, respiratory and behavioral analysis”.

2) We interpret the “pause” period as a delay to initiate the subsequent breath and this implicates the preBötC in OIRD. This falls in line with the fact that the autonomous respiratory rhythm generated by the preBötC slice slows after bath administration of opioid agonists, independent of any other brain areas. Further, after deletion of Oprm1 from the preBötC pause lengths are significantly decreased (Figure 2J). Since our recordings occur in hypercapnia, the pause terminates with a brief active expiration. However, active expiration depends upon preBötC activity (Huckstepp et al., 2016) and thus remains consistent with the pause reflecting a delay in preBötC inspiratory initiation.

DiscussionThe authors do not discuss previous studies about MOR-/-, preBötC, etc. There are many studies investigating this topic that could be discussed in the Discussion. Dahan et al., 2001, Montandon et al., 2011, 2016, Varga et al., 2020, Stucke et al., Mustapic et al., Lalley et al. etc.

We appreciate your recommendations for citations. We have already discussed the importance of and cited Montandon et al., 2011, Varga et al., 2020 and Mustapic et al. in our manuscript.